# Clinical Safety of Expanded Hemodialysis Compared with Hemodialysis Using High-Flux Dialyzer during a Three-Year Cohort

**DOI:** 10.3390/jcm11082261

**Published:** 2022-04-18

**Authors:** Nam-Jun Cho, Seung-Hyun Jeong, Ka Young Lee, Jin Young Yu, Samel Park, Eun Young Lee, Hyo-Wook Gil

**Affiliations:** Department of Internal Medicine, Soonchunhyang University Cheonan Hospital, Cheonan 31151, Korea; chonj@schmc.ac.kr (N.-J.C.); wbsw611@naver.com (S.-H.J.); kayoung47@nate.com (K.Y.L.); yujjinyoung@gmail.com (J.Y.Y.); samelpark17@schmc.ac.kr (S.P.); eylee@schmc.ac.kr (E.Y.L.)

**Keywords:** cytokines, interleukin-6, expanded hemodialysis

## Abstract

Expanded hemodialysis (HD) equipped with a medium cut-off (MCO) membrane provides superior removal of larger middle molecules. However, there is still little research on the long-term benefits of expanded HD. Over a three-year period, this observational study evaluated the efficacy and safety profile of expanded HD for inflammatory cytokines, including IL-6. We conducted a prospective cohort study to investigate the inflammatory cytokine changes and a retrospective observational cohort study to investigate long-term clinical efficacy and safety over a three-year period. We categorized the patients according to dialyzer used: MCO and high-flux (HF) dialyzer. The inflammatory cytokines, including IFN-γ, IL-1β, IL-6, and TNF-α, were measured annually. The concentrations and changes of the four cytokines over time did not differ between the HF group (*n* = 15) and MCO group (*n* = 27). In both prospective and retrospective (HF group, *n* = 38; MCO group, *n* = 76) cohorts, there were no significant differences in either death, cardiovascular events, infections, or hospitalizations. Furthermore, the temporal changes in laboratory values, including serum albumin and erythropoietin prescriptions, did not differ significantly between the two groups in either the prospective or retrospective cohorts. In conclusion, clinical efficacy and safety outcomes, as well as inflammatory cytokines, did not differ with expanded HD compared with HF dialysis during a three-year treatment course, although the level of inflammatory cytokine was stable.

## 1. Introduction

Uremic toxins accumulate in body fluids during the course of progressive chronic kidney disease. Among uremic toxins, larger middle molecules are associated with inflammation and cardiovascular events, as well as other dialysis-related comorbidities [1]. Hemodialysis (HD) with high-flux (HF) dialyzers can remove small conventional middle molecules (0.5–25 kD) such as beta-2 microglobulin. However, clearance of larger middle molecules (>25 kD) needs either an HD with a higher permeable dialyzer, e.g., expanded HD, or convection dialysis, e.g., online hemodiafiltration. Expanded HD is a treatment in which diffusion and convection are integrated inside a dialyzer equipped with a medium cut-off (MCO) membrane [2]. An MCO membrane (Theranova^®^ dialyzer) is a hollow-fiber, single-use dialyzer, with improved removal of large proteins (up to 25 kD), as well as selective maintenance of essential proteins such as albumin [3]. Recently, some studies have shown that expanded HD provides superior removal of larger middle molecules, as exemplified by free light chains, compared to a similar size HF dialyzer, while maintaining serum albumin levels [4,5,6,7,8]. However, the prior studies examined the clinical safety of expanded HD with a limited follow-up duration. Our group had previously published a study about the one-year effects of expanded HD on middle uremic toxins [9]. We showed that expanded HD provides an excellent reduction rate of lambda and kappa free light chains, but long-term change was not significant. We suggested that expanded HD can be used effectively and safely in a conventional HD setting. However, there is still little research on the long-term benefits of using expanded HD. Furthermore, it should also be noted that the reduction rate of free light chains could have an effect on cardiovascular events, inflammation, and cytokines, regardless of serum level. Middle uremic toxins have been predominantly involved in inflammation, immune response, and cardiovascular disease in patients with end-stage renal disease [1,10]. Among middle molecules, inflammatory cytokines are suspected as being a culprit of cardiovascular comorbidity in hemodialysis. The uremic state probably increases the peripheral cell release of proinflammatory cytokines, e.g., interleukin (IL)-1β, IL-6, tumor necrosis factor (TNF)-α, etc., and slows removal of the latter, resulting in a net increase [11,12,13,14]. Short-term implementation of expanded HD to reduce inflammatory cytokines, including IL-6 [2,6,15], is controversial. Long-term effects of expanded HD on cytokines is also still unclear. Therefore, this three-year observational study was conducted to evaluate the efficacy and clinical safety profile of expanded HD against inflammatory cytokines, including IL-6.

## 2. Materials and Methods

### 2.1. Study Population and Study Design

This study included patients who received maintenance HD at the Dialysis Unit of Soonchunhyang University Cheonan Hospital in Cheonan, Republic of Korea, in October 2017. Firstly, we designed an open-label nonrandomized prospective cohort study to investigate the long-term cytokine changes according to the dialyzers used. The design of this prospective study was registered with the Clinical Research Information Service (CRiS) at the Korea Centers for Disease Control and Prevention (KCT0005557, registration date: 2 September 2020). Secondly, we executed a retrospective observational cohort study to investigate the long-term clinical efficacy and safety of expanded hemodialysis. Given that this retrospective cohort included all hemodialysis patients at the dialysis facility, the retrospective cohort included the patients of the prospective study as well as additional patients.

The two cohorts shared identical inclusion and exclusion criteria. Inclusion criteria included age greater than 20 years and ESRD requiring HD treatment thrice weekly for at least six months. Exclusion criteria were patients who were not clinically stable, had a hemolytic disease, or had a history of blood transfusion during the study period. Patients were followed for three years, from October 2017 to January 2021. In most analyses, patients who were followed up for the entire period were included, but patients who dropped out were also included in the survival analysis. We did not alter the patient dialysis regimen, including blood flow, dialysate flow, and treatment time duration per session. All patients were treated with bicarbonate dialysate of ultrapure quality. The dialyzers were used only once.

These studies were conducted in accordance with the ethical principles of the Declaration of Helsinki. The Institutional Review Board of Soonchunhyang University Cheonan Hospital approved the study protocol (2018-07-033, 2020-08-022). All study participants of the prospective cohort provided informed written consent before study enrollment, and informed consent was waived in the retrospective cohort.

### 2.2. Allocation of the Dialysis Membranes

In the dialysis facility, three different kinds of dialysis machines, including Artis (Baxter, Chicago, IL, USA), AK200 ULTRA S (Baxter, Chicago, IL, USA), and Fresenius Medical Care (FMC) 5008 dialysis machines (FMC Deutschland, Bad Homburg, Germany) were used. The machine assigned to the patients was the machine available at the first HD in this facility and was not changed thereafter. The membrane selected for use was either Baxter’s (Revaclear 400 or Theranova 400) or FMC’s (FX CorDiax 80), according to the particular dialysis machine used by the patient. Before the Theranova 400 dialyzer was introduced to the facility (October 2017), all patients were on HD with HF membranes (Revaclear 400 and FX CorDiax 80). After the introduction of the Theranova 400 dialyzer, patients who used the FX CorDiax 80 continued to use it, but the Revaclear 400 membrane was changed to an MCO membrane (Theranova 400). Therefore, there has been an HF group (FX CorDiax 80) and an MCO group (Theranova 400) in the facility since October 2017.

### 2.3. Samplings and Analyses

Baseline predialysis blood samples were drawn before the introduction of the MCO membrane, and predialysis samples had been drawn yearly following baseline. Blood was collected in serum-separating tubes and allowed to stand for 30 min, then centrifuged for 10 min at 3000 rpm at room temperature within one hour. The serum was extracted and stored at −70 °C until analysis. The levels of serum interferon (INF)-γ, IL-1β, IL-6, and TNF-α were measured using a 32 × 4 multiplex custom-designed plate on an ELLA™ automated immunoassay system (Bio-Techne, San Jose, CA, USA) according to the manufacturer’s instructions.

### 2.4. Clinical Outcomes

We set four clinical events for survival analysis, including death, cardiovascular events, infectious events, and hospitalizations. A cardiovascular event was defined as the combination of myocardial infarction, stroke, hospitalization because of heart failure, or revascularization, including percutaneous coronary intervention and coronary artery bypass graft. An infection event was defined as an episode characterized by clinical signs and symptoms necessitating antibiotic therapy. Hospitalization was defined as any hospital admission resulting in inpatient status for 24 h or longer.

We investigated changes in laboratory findings, including hemoglobin, platelet, transferrin saturation, serum albumin, total calcium, phosphorus, intact parathyroid hormone, and alkaline phosphatase. The average laboratory values were calculated at three-month intervals. We also investigated the prescriptions of erythropoietin-stimulating agents (ESA), phosphate binders, cinacalcet, and paricalcitol over a three-month interval. Given that darbepoetin is the most commonly used ESA in this dialysis facility, we calculated monthly ESA usage based on it. The prescription amount of phosphate binder and cinacalcet was calculated as the monthly average number of tablets.

### 2.5. Statistical Analysis

Statistical analyses were performed using R version 4.0.3 (The R Foundation for Statistical Computing, Vienna, Austria). Categorical variables are expressed as counts (percentage), normally distributed continuous variables as means ± SD, and non-normally distributed continuous variables as medians (interquartile ranges). Differences between two independent groups were analyzed by Student’s *t*-tests for normally distributed continuous variables, and by Mann–Whitney U test for non-normally distributed continuous variables. Differences between two dependent groups were analyzed by paired samples *t*-tests for normally distributed continuous variables and by Wilcoxon signed-rank test for non-normally distributed continuous variables. Categorical variables were analyzed using the Pearson’s chi-squared test or Fisher’s exact test, as appropriate.

We used a linear mixed model to determine the relationship between the membrane used and the sustained outcome over time. Fixed effects included the membranes, age, gender, dry weight, and presence of diabetes. Random effects for intercepts and slopes were included in all mixed models. We focused on the interaction between time and membrane type, revealing the temporal differences in laboratory values between MCO and high-flux groups. Satterthwaite’s method was used to approximate the degrees of freedom and statistical significance of linear mixed models.

As appropriate, values of variables with skewed distributions were log transformed or square-root transformed. *p*-values < 0.05 were regarded as statistically significant, and two-tailed tests were performed for all hypothesis tests.

## 3. Results

### 3.1. Characteristics of Study Subjects

A total of 139 patients (HF group, 55 patients; MCO group, 84 patients) were enrolled in the retrospective cohort study at baseline. After excluding 25 patients who met exclusion criteria, 114 patients (HF group, 38 patients; MCO group, 76 patients) were selected for survival analyses of clinical events. We followed up 76 patients during the entire study period to analyze changes in laboratory findings and drug dosages. In the prospective cohort, a total of 57 patients were enrolled. After excluding those patients who dropped out during the study, we selected 42 patients (HF group, 15 patients; MCO group, 27 patients) for cytokine analysis.

The prospective and retrospective cohorts showed no significant differences in baseline characteristics between the HF and MCO groups (Table 1). The mean age was 53.5 ± 10.6 years. Twenty-three (54.8%) patients were male, and the median dialysis vintage in the prospective cohort was 77 (38–139) months. In the retrospective cohort, the mean age of the 66 (57.9%) male patients was 54.8 ± 12.6 years, and the median dialysis vintage was 73 (40–140) months.

### 3.2. Clinical Events during the Three-Year Treatment Period

We investigated the clinical outcomes, including all-cause death, cardiovascular events, infections, and hospitalizations, in the retrospective cohort according to the membrane used. During the three-year study period, there were five deaths (two in the HF group and three in the MCO group, *p* = 0.74), nine CV events (three in the HF group and six in the MCO group, *p* = 0.97), 25 infection events (nine in the HF group and 16 in the MCO group, *p* = 0.74), and 55 hospitalizations (17 in the HF group and 38 in the MCO group, *p* = 0.63). There were no significant differences between the two groups in the occurrences of these four clinical events (Figure 1).

### 3.3. Serial Changes of Laboratory Findings during the Three-Year Treatment Period

Figure 2 shows important laboratory parameter changes over the three years of treatment. Serum albumin levels were lower in the MCO group as compared to the HF group (mean difference = −0.180, *p* = 0.001), but they did not differ over time (mean changes difference per month = −0.0003, *p* = 0.855). Serum calcium levels decreased over time (mean changes per month = −0.012, *p* < 0.001), and intact PTH levels increased over time in both groups (mean changes per month = 0.012, *p* = 0.001). However, there were no significant differences in serum calcium and intact PTH changes over time between the HF and MCO groups (mean changes difference of serum calcium per month = −0.004, *p* = 0.320; mean changes difference of intact PTH per month = 0.183, *p* = 0.316). The changes of other laboratory parameters, likewise, did not differ between the two groups (Appendix A).

### 3.4. Serial Changes of Medication Prescriptions during the Three-Year Treatment Period

The monthly dose of certain drugs and their changes over the study periods are presented in Figure 3. The drugs used for analyses were darbepoetin, phosphate binders, cinacalcet, and paricalcitol. The mean erythropoietin-stimulating agent (darbepoetin) doses and their mean changes over time did not differ between the high-flux group and MCO group (mean difference = −0.102, *p* = 0.408; mean changes difference per month = 0.013, *p* = 0.258). Likewise, phosphate binders, cinacalcet, and paricalcitol showed no differences in mean doses or mean changes between the two groups over time (Appendix A).

### 3.5. Serial Changes of Inflammatory Cytokines during the Three-Year Treatment Period

The inflammatory cytokines, including IFN-γ, IL-1β, IL-6, and TNF-α, in the prospective cohort were measured annually. The changes in the inflammatory cytokines during the study period are presented in Figure 4. The concentrations decreased over time in IFN-γ (mean changes per year = −0.262, *p* < 0.001) and IL-1β (mean changes per year = −0.369, *p* = 0.001) (Appendix A). Each annual concentration of IL-1β, IL-6, and TNF-α is presented in Appendix A. The IFN-γ values were especially low in the third year (baseline, 1.27 ± 0.60 pg/mL; first year, 1.16 ± 0.85 pg/mL; second year, 1.15 ± 0.80 pg/mL; third year, 0.48 ± 0.49 pg/mL). The concentrations of these four cytokines did not differ between the HF group and the MCO group, and the concentration changes over time also did not differ between the two (Appendix A).

## 4. Discussion

During the three-year treatment period, our study found no differences between the two groups in clinical events or laboratory findings, including albumin or medication usage. Interestingly, in both groups, inflammatory cytokines were stable over the three years and the IFN-γ and IL-Iβ decreased over the three years.

Expanded HD could remove the middle uremic toxins between 15 and 60 kDa, which include inflammatory cytokines [13,16]. However, the effectiveness of expanded HD in clearing cytokines has not been well documented, and it is unknown whether prolonged HD can lower cytokine levels.

IL-6, an inflammatory cytokine, is one of the most highly regulated mediators of inflammation (increasing from 1–5 pg/mL to several µg/mL in certain conditions) and plays a central role in various diseases [17,18,19]. IL-6 and soluble IL-6 receptors (sIL-6R) are considered important prognostic markers of clinical outcomes in ESRD patients [19]. High IL-6 levels contribute to dialysis-associated malnutrition and are prognostic of cardiovascular risk, which is an adverse outcome of hemodialysis [12,19,20,21]. Raised serum IL-6 and sIL-6R levels at the beginning of treatment remain powerful predictors of mortality in HD patients [12]. Clinical studies on whether expanded HD can effectively remove IL-6 are controversial. 

According to Zickler D et al., at four weeks, the gene expression of TNF-α and IL-6 in peripheral blood mononuclear cells with the MCO dialyzer was reduced to a significantly greater extent than with the HF dialyzer [2]. Expanded HD has a greater reduction rate of TNF-α at 24 weeks but not for IL-6 as compared to HD with HF [4]. However, the changes in IL-6 and TNF-α levels at baseline before dialysis did not significantly differ [4]. 

In our study, expanded HD did not reduce serum concentrations of IL-6 in HD patients over a three-year period. Compared to HF groups, expanded HD was not more effective in cytokine clearance. We think that the inflammatory cytokines do not increase during HD with MCO or HF membranes because the biocompatibility of the dialysis membranes is excellent. Recent technological advances have improved the biocompatibility of dialyzer membranes [20,21]. Given that the enrolled subjects were stable during the treatment period, it is thought that the inflammatory cytokines showed little change, and even IFN-γ and IL-1β gradually decreased over the three years of treatment.

Cytokine reductions during expanded HD can be subtle because baseline IL-6 levels are low in HD patients. In our study, the mean IL-6 was 5.70 pg/mL, which is a similar baseline concentration to previous studies [13]. One observational study showed that expanded hemodialysis improves survival and effectively removes cytokines in patients with COVID-19 [22]. Further studies should reveal whether expanded HD could effectively remove inflammatory cytokines in patients with severe infection.

A recent meta-analysis study showed that expanded HD removed approximately 2 g of albumin per 4 h conventional hemodialysis session, resulting in a decreased serum albumin level of 1.2 g/dL over the short term (<24 weeks), which returned to baseline thereafter [15]. In our study, serum albumin was lower at baseline in the MOC group as compared to the HF group. However, serum albumin did not change during the first six months. Moreover, the variation of albumin did not differ significantly over the three-year treatment period. Recently, expanded HD reduced ESA resistance as compared to HD with HF [23]. However, two other studies showed that expanded HD had no effect on ESA dosage [24,25]. Our study showed that expanded HD did not reduce the ESA dose as compared to HF HD. We think that the inflammatory statuses were similar in both groups, which may have influenced the resistance of EPO in patients with HD. In addition, mineral bone disease biomarkers and dialysis efficiency were similar in both groups. Meta-analysis studies have also shown that MCO improves clinical outcomes, such as a reduction in symptom burdens and infections, recovery time, and length of hospital stay, and quality of life [23]. The MCO membrane did not improve such clinical outcomes as all-cause death, cardiovascular events, infections, or hospitalizations as compared to the HF group. Several factors other than the dialysis membrane are related to the clinical outcomes in patients with hemodialysis [26]. We think that the effect of MCO membranes was not superior to HF membranes in our study because all subjects in both groups were well managed.

In our study, two different materials of dialyzers were used. The HF group used a polysulfone membrane and the MCO group used a polyarylethersulfone and polyvinylpyrrolidone blend membrane. A previous study showed that these two dialyzers have similar biocompatibility [27]. Our study also supported that recently used synthetic membranes have very stable biocompatibility.

Our study has some limitations. Firstly, this study was not randomized. Secondly, our cohort study was at a single center. Thirdly, the number of patients included in the cytokine analysis was not big, so there was a limitation for catching subtle differences. However, our study has the advantage of observing changes in inflammatory cytokines over the long period of three years.

## 5. Conclusions

Inflammatory cytokines were not lower in expanded HD as compared to HF during a three-year treatment period, although the level of inflammatory cytokines was stable. This can explain the biocompatibility of the expanded HD. In addition, there were no significant differences in clinical efficacy and safety outcomes between expanded HD and HF dialysis.

## Figures and Tables

**Figure 1 jcm-11-02261-f001:**
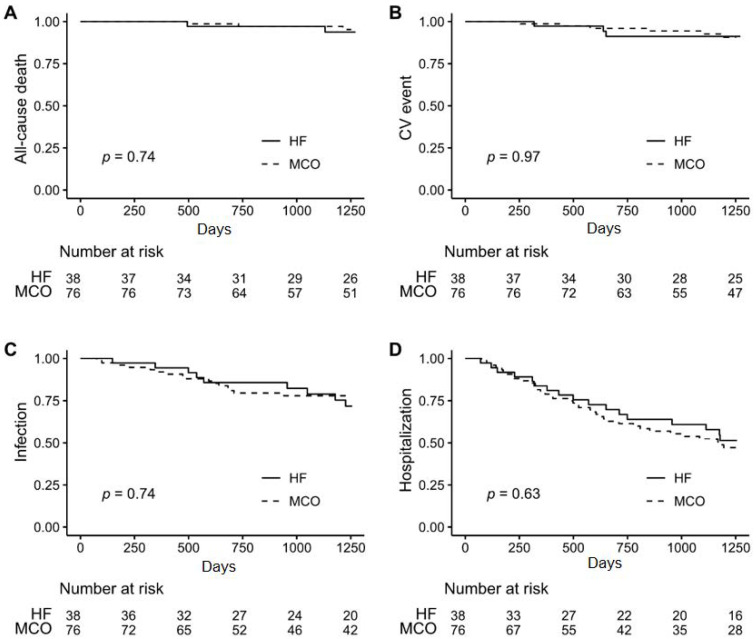
Clinical events occurrence according to the hemodialysis membrane types. Clinical outcomes included all-cause death (**A**), cardiovascular events (**B**), infectious events (**C**), and hospitalizations (**D**). HF, high-flux membrane; MCO, medium cut-off membrane; CV, cardiovascular.

**Figure 2 jcm-11-02261-f002:**
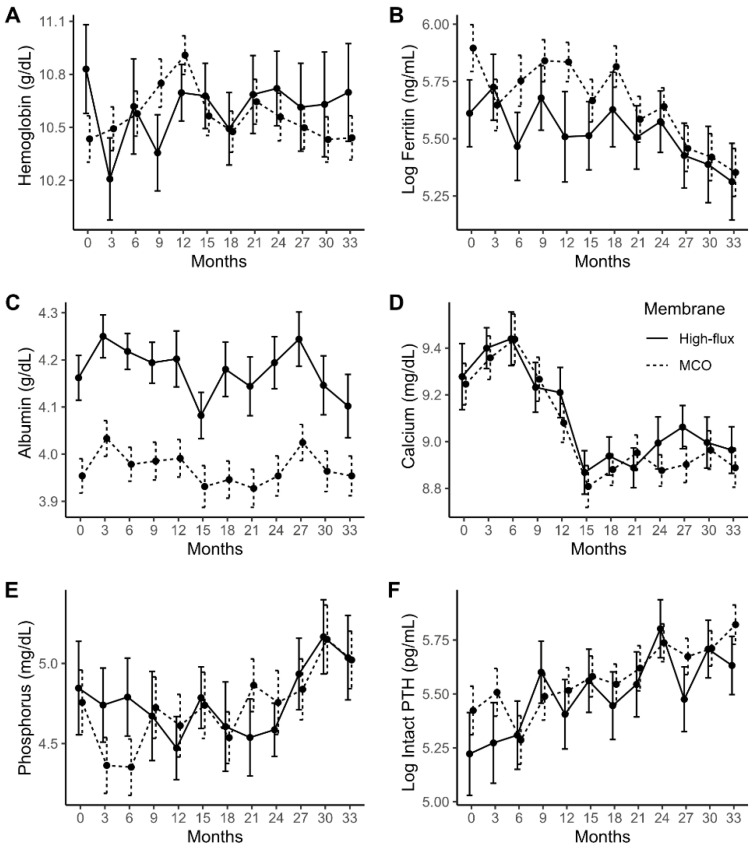
Changes of laboratory findings during the three-year treatment period according to the membrane type. The average values of the laboratory parameters including hemoglobin (**A**), ferritin (**B**), serum albumin (**C**), total calcium (**D**), phosphorus (**E**), and intact PTH (**F**) are shown. Data are presented as arithmetic means (black dots) and standard errors (error bars) as an error bar plot. The values of the high-flux group are represented as solid lines, and those of the MCO group as dash lines. Ferritin and intact PTH values were log transformed. MCO, medium cut-off membrane; PTH, parathyroid hormone.

**Figure 3 jcm-11-02261-f003:**
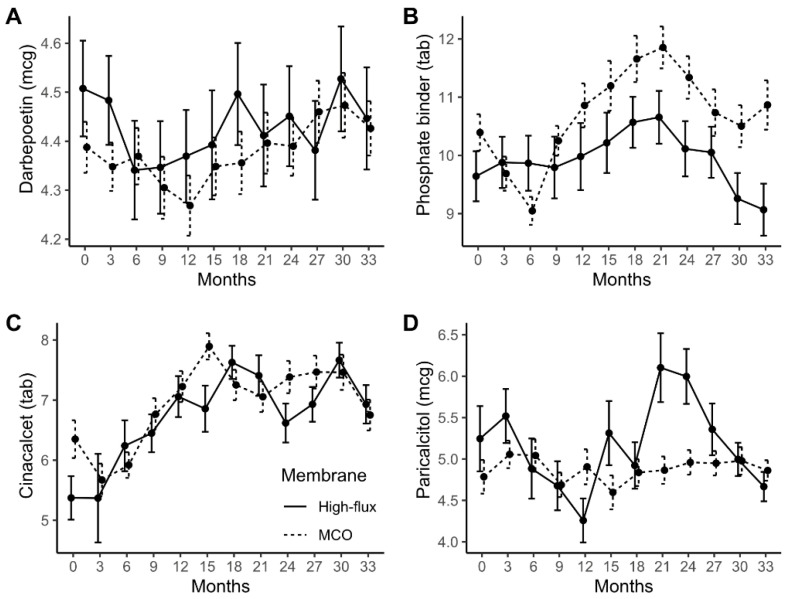
Changes in medication usage during the three-year treatment period according to the membrane type. The average monthly doses of darbepoetin (**A**), phosphate binders (**B**), cinacalcet (**C**), and paricalcitol (**D**) are shown. Data are presented as arithmetic means (black dots) and standard errors (error bars) as an error bar plot. The darbepoetin and paricalcitol doses are expressed in microgram units. The amounts of phosphate binders and cinacalcet are expressed in tablet numbers. The values of the high-flux group are represented as solid lines and those of the MCO group as dash lines. Darbepoetin doses were log transformed. The quantities of phosphate binder, cinacalcet, and paricalcitol were square-root transformed. Phosphate binder included calcium carbonate, calcium acetate, and sevelamer. MCO, medium cut-off membrane.

**Figure 4 jcm-11-02261-f004:**
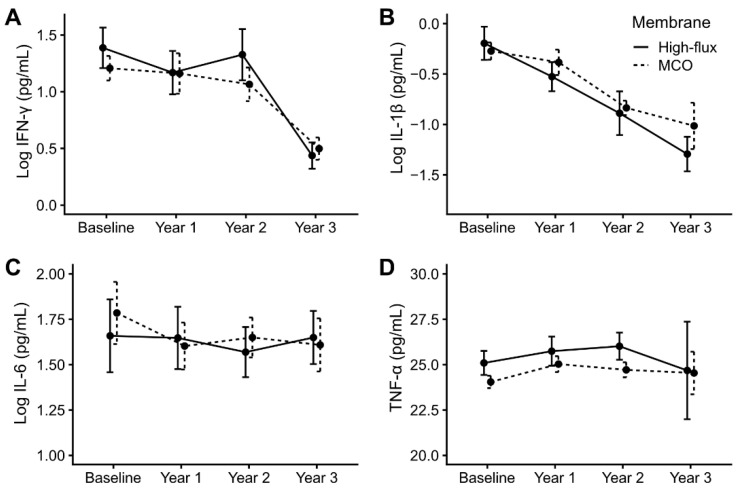
Changes in cytokine concentrations during the three-year treatment period according to the membrane type. The mean values of cytokines during the study periods are presented: (**A**) interferon-γ, (**B**) interleukin-1β, (**C**) interleukin-6, (**D**) tumor necrosis factor-α. Data are presented as arithmetic means and standard errors as an error bar plot. The values of interferon-γ, interleukin-1β, and interleukin-6 were logarithmically transformed. The values of the high-flux group are represented as solid lines and those of the MCO group as dash lines. MCO, medium cut-off; IFN interferon; IL, interleukin; TNF, tumor necrosis factor.

**Table 1 jcm-11-02261-t001:** Cohort demographics and clinical characteristics.

	Prospective Cohort (*n* = 42)			Retrospective Cohort (*n* = 114)		
	HF Group (*n* = 15)	MCO Group (*n* = 27)	*p*-Value	HF Group (*n* = 38)	MCO Group (*n* = 76)	*p*-Value
Age, years	56.7 ± 11.7	51.7 ± 10.3	0.160	54.7 ± 11.7	54.8 ± 13.1	0.967
Gender, *n* (%)			0.405			0.070
Male	10 (66.7)	13 (48.1)		27 (71.1)	39 (51.3)	
Female	5 (33.3)	14 (51.9)		11 (28.9)	37 (48.7)	
Height, cm	165.5 ± 6.7	162.2 ± 9.0	0.228	165.0 ± 7.7	162.1 ± 9.7	0.120
Dry weight, kg	58.2 ± 11.8	58.1 ± 12.5	0.984	59.9 ± 12.5	60.1 ± 14.1	0.953
ESRD cause, *n* (%)			0.083			0.087
Diabetic nephropathy	5 (33.3)	6 (22.2)		11 (28.9)	25 (32.9)	
Hypertensive	5 (33.3)	7 (25.9)		11 (28.9)	14 (18.4)	
Glomerulonephritis	3 (20.0)	14 (51.9)		10 (26.3)	33 (43.4)	
Others	2 (13.3)	0 (0.0)		6 (15.8)	4 (5.3)	
Dialysis vintage, months	131 (35, 214)	56 (45, 119)	0.376	99 (29, 203)	68 (46, 126)	0.643
Hypertension, present	15 (100.0)	25 (92.6)	0.530	36 (94.7)	71 (93.4)	1.000
Diabetes, present	5 (33.3)	10 (37.0)	1.000	11 (28.9)	30 (39.5)	0.370
Heart disease, present	3 (20.0)	2 (7.4)	0.329	5 (13.2)	10 (13.2)	1.000
Vascular access, *n* (%)			1.000			0.917
Native AV fistula	13 (86.7)	24 (88.9)		33 (86.8)	68 (89.5)	
PTFE graft	2 (13.3)	3 (11.1)		5 (13.2)	8 (10.5)	
Hemoglobin, g/dL	10.5 ± 1.5	10.1 ± 1.6	0.473	10.3 ± 1.5	10.4 ± 1.4	0.761
Albumin, g/dL	4.11 ± 0.24	3.99 ± 0.34	0.237	4.05 ± 0.32	3.94 ± 0.35	0.093
Urea nitrogen, mg/dL	65.6 ± 16.3	62.7 ± 20.3	0.633	64.0 ± 19.4	58.1 ± 16.8	0.101
Creatinine, mg/dL	9.72 ± 3.50	9.64 ± 2.90	0.940	9.63 ± 3.04	9.59 ± 2.72	0.947
Triglycerides, mg/dL	92 (69, 122)	90 (57, 137)	0.793	96 (64, 149)	95 (59, 133)	0.477
Total cholesterol, mg/dL	134.1 ± 32.4	146.0 ± 29.9	0.238	135.8 ± 31.0	141.0 ± 30.7	0.397
Phosphorus, mg/dL	4.95 ± 1.70	4.81 ± 1.65	0.787	4.69 ± 1.48	4.53 ± 1.67	0.614
Calcium, mg/dL	9.07 ± 0.56	9.44 ± 0.67	0.082	9.28 ± 0.69	9.33 ± 0.74	0.757
Ferritin, ng/mL	268 (129, 363)	290 (227, 437)	0.323	269 (153, 385)	252 (139, 419)	0.995
Intact PTH, pg/mL	164 (99, 309)	205 (96, 298)	0.795	152 (78, 281)	185 (106, 286)	0.260
Kt/V per session	1.90 ± 0.37	1.89 ± 0.37	0.917	1.82 ± 0.33	1.87 ± 0.38	0.475
URR, %	0.78 ± 0.06	0.77 ± 0.07	0.801	0.77 ± 0.06	0.77 ± 0.07	0.827

Data are presented as mean ± SD, median (interquartile range), or count (%) as appropriate. AV, arteriovenous; HF, high-flux membrane; MCO, medium cut-off membrane; ESRD, end-stage renal disease; PTFE, polytetrafluoroethylene; PTH, parathyroid hormone; URR, urea reduction ratio.

## Data Availability

The datasets used and/or analyzed during the current study are available from the corresponding author upon reasonable request.

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
