# Peer review of "Clinical Safety of Expanded Hemodialysis Compared with Hemodialysis Using High-Flux Dialyzer during a Three-Year Cohort"

_jcm, 2022, doi:10.3390/jcm11082261_

Round 1
Reviewer 1 Report
The authors have described the biocompatibility of both expanded hemodialysis (HD) and HD using high flux (HF) membrane by measuring inflammatory cytokines. Two different membranes, i.e., polyarylethersulfone (PAES) and polysulfone (PS) were used in this study.
Biocompatibility in HD is not only associated with inflammatory cytokines but the dialysis membrane materials also affect patients undergoing HD. Please describe the possibility of the dialysis membrane material's effect.
In addition, what about the difference between solute removal efficacy e.g., removal rate or sieving coefficient of β2-microglobulin and/or cytokines for both PAES and PS? This information will be required to compare both expanded HD and HF treatment to clarify the differences in background in this study.
Author Response
The authors have described the biocompatibility of both expanded hemodialysis (HD) and HD using high flux (HF) membrane by measuring inflammatory cytokines. Two different membranes, i.e., polyarylethersulfone (PAES) and polysulfone (PS) were used in this study.
Comment 1. Biocompatibility in HD is not only associated with inflammatory cytokines but the dialysis membrane materials also affect patients undergoing HD. Please describe the possibility of the dialysis membrane material's effect.
Response: In our study, two membranes included polysulfone(Helixone) or polyarylethersulfone and Polyvinylpyrrolidone blend membrane (Theranova). Previous study showed that the membrane using polyarylethersulfone in conjunction with Polyvinylpyrrolidone has complement-activation potential and neutropenia similar to Fresenius Polysulfone (Am J Kidney Dis. 2000 Aug;36(2):345-52). Recently used synthetic membranes (polyarylethersulfone and polysulfone PS) have very stable biocompatibility. However some hypersensitivity reactions related to PS have been reported (Ann Allergy Asthma Immunol . 2022 Mar 12;S1081-1206(22)00175-2). So we insert these backgrounds in the discussion section as follows:
“In our study, two different materials of dialyzers were used. HF group used polysulfone membrane, and MCO group used polyarylethersulfone and polyvinylpyrrolidone blend membrane. Previous study showed that these two dialyzers have similar biocompatibility. Our study also supported that recently used synthetic membranes have very stable biocompatibility.”
Comment 2. In addition, what about the difference between solute removal efficacy e.g., removal rate or sieving coefficient of β2-microglobulin and/or cytokines for both PAES and PS? This information will be required to compare both expanded HD and HF treatment to clarify the differences in background in this study.
Response :
|
QB 200 / QD 500 ml/min |
FX80 |
Theranova |
Revaclear |
|
Sieving Coefficient |
|
|
|
|
B2MG |
0.9 |
1.0 |
0.95 |
|
Myoglobin |
0.5 |
0.9 |
0.68 |
Membrane's ability to remove uremic toxin is likely due to structural differences rather than membrane material differences. Theravnoa and revaclear are composed with the same materials (polyarylethersulfone and polyvinylpyrrolidone blend membrane). However, expanded HD have superior ability to remove large middle molecules because Theranova dialyzer (expanded HD) have an unique cut-off and high retention onset profile permit filtration. So we did not describe the sieving coefficient according to materials of dialyzers.
Reviewer 2 Report
In this monocentric work the authors compare the use of two dialytic methods (using high-flux and medium cut-off membranes) evaluating their effects from a clinical and laboratory point of view, focusing on pro-inflammatory cytokines. Although not novel in the design, the work seems well drawn, with however a not large population for the prospective analysis. The authors found no difference between the two treatments for inflammatory cytokines, laboratory clinical and hard outcomes. There are, however, some things to report:
Major revisions:
- the title of the work stresses the importance of the analysis of inflammatory cytokines, underlining the focus of the paper. However, the group of patients in which cytokines are studied is small; in addition, samples were taken only once a year, all this possibly underestimating the biological complexity of the underlying system. However, the values of the two groups are comparable. In this sense perhaps I would modify the title by including in it the general clinical aspects and the long follow-up.
- In line with the above mentioned, in the conclusion of the abstract I would stress in a more general way the possible similarity for the considered parameters between the two treatments, thus also giving importance to the other considered features.
- I would add in the limitations the number of sample
- In view of the substantial comparability of the results of the two methods, the authors could therefore point out any cost differences in the discussions
Minor revisions:
-Include information on how many patients were on triple-weekly, bi-weekly and single-weekly dialysis, since this better define the population;
- Materials and methods: line 105-109, typing error
Author Response
In this monocentric work the authors compare the use of two dialytic methods (using high-flux and medium cut-off membranes) evaluating their effects from a clinical and laboratory point of view, focusing on pro-inflammatory cytokines. Although not novel in the design, the work seems well drawn, with however a not large population for the prospective analysis. The authors found no difference between the two treatments for inflammatory cytokines, laboratory clinical and hard outcomes. There are, however, some things to report:
Major revisions:
Comment 1. the title of the work stresses the importance of the analysis of inflammatory cytokines, underlining the focus of the paper. However, the group of patients in which cytokines are studied is small; in addition, samples were taken only once a year, all this possibly underestimating the biological complexity of the underlying system. However, the values of the two groups are comparable. In this sense perhaps I would modify the title by including in it the general clinical aspects and the long follow-up.
Response: Thank you for the good comment.
We changed the title as follows:
Clinical safety of Expanded Hemodialysis compared with Hemodialysis using High flux dialyzer during a Three Year Cohort
Comment 2. In line with the above mentioned, in the conclusion of the abstract I would stress in a more general way the possible similarity for the considered parameters between the two treatments, thus also giving importance to the other considered features.
Response: Thank you for your good comment. We changed the conclusion in abstract as follows:
In conclusion, clinical efficacy and safety outcomes, as well as inflammatory cytokines, did not differ with expanded HD compared with HF dialysis during a three-year treatment course, although the level of inflammatory cytokine was stable.
Comment 3. I would add in the limitations the number of sample
Response: We added the sample size in the limitation section.
: Third, the number of patients included in the cytokine analysis was not big, so there was a limitation for catching subtle differences.
Comment 4. In view of the substantial comparability of the results of the two methods, the authors could therefore point out any cost differences in the discussions
Response: We did not analyze the cost differences between two groups, because the cost is determined according to each country's medical policy. Recent, Colombia study showed expanded HD was statistically significantly associated with reduced hospitalization days and lower medication dosages (Ther Apher Dial. 2021 Oct;25(5):621-627.). However, clinical outcomes including hospitalization and erythropoietin dosage were not different in our study. Therefore, we did not include the cost difference between the two groups in this study because it was not meaningful.
Minor revisions:
Comment 5. Include information on how many patients were on triple-weekly, bi-weekly and single-weekly dialysis, since this better define the population;
Response: At enrollment, all patients received HD treatment thrice weekly for at least six months. We already described these in the Material and Methods section (Study Population and Study Design) as follows: “Inclusion criteria included age greater than 20 years and ESRD requiring HD treatment thrice weekly for at least six months.”
Comment 6. Materials and methods: line 105-109, typing error
Response: at -70 ℃ à at -70℃